# Reset-free Reinforcement Learning with World Models

**Zhao Yang**                                                                                   *z.yang@liacs.leidenuniv.nl*
*The Leiden Institute of Advanced Computer Science*
*Leiden University*

**Thomas M. Moerland**
*The Leiden Institute of Advanced Computer Science*
*Leiden University*

**Mike Preuss**
*The Leiden Institute of Advanced Computer Science*
*Leiden University*

**Aske Plaat**
*The Leiden Institute of Advanced Computer Science*
*Leiden University*

**Edward S. Hu**
*GRASP Lab*
*University of Pennsylvania*

**Reviewed on OpenReview:** *https://openreview.net/forum?id=ZdMIXltJzK*

## Abstract

Reinforcement learning (RL) is an appealing paradigm for training intelligent agents, enabling policy acquisition from the agent's own autonomously acquired experience. However, the training process of RL is far from automatic, requiring extensive human effort to reset the agent and environments. To tackle the challenging reset-free setting, we first demonstrate the superiority of model-based (MB) RL methods in such setting, showing that a straightforward adaptation of MBRL can outperform all the prior state-of-the-art methods while requiring less supervision. We then identify limitations inherent to this direct extension and propose a solution called model-based reset-free (MoReFree) agent, which further enhances the performance. MoReFree adapts two key mechanisms, exploration and policy learning, to handle reset-free tasks by prioritizing task-relevant states. It exhibits superior data-efficiency across various reset-free tasks without access to environmental reward or demonstrations while significantly outperforming privileged baselines that require supervision. Our findings suggest model-based methods hold significant promise for reducing human effort in RL. Website: `https://yangzhao-666.github.io/morefree`

## 1 Introduction

Reinforcement learning presents an attractive framework for training capable agents. At first glance, RL training appears intuitive and autonomous - once a reward is defined, the agent learns from its own automatically gathered experience. However, in practice, RL training often assumes the access to environmental resets that can require significant human effort to setup, which poses a significant barrier for real world applications of RL like robotics.

Most RL systems on real robots to date have employed various strategies to implement resets, all requiring a considerable amount of effort (Levine et al., 2016; Yahya et al., 2017; Zhu et al., 2019; Nagabandi et al., 2020). In Nagabandi et al. (2020), which trains a dexterous hand to rotate balls, the practitioners had to

(1) position a funnel underneath the hand to catch dropped balls, and (2) deploy a separate robot arm to pick up the dropped balls for resets, and (3) script the reset behavior. These illustrate that even for simple behaviors, proper implementation of reset mechanisms can result in significant human effort and time.

Rather than depending on human-engineered reset mechanisms, the agent can operate within a reset-free training scheme, learning to reset itself (Eysenbach et al., 2017; Sharma et al., 2021a; 2022; Haldar et al., 2023) or train a policy capable of starting from diverse starting states (Zhu et al., 2020). However, the absence of resets introduces unique exploration challenges. Without periodic resets, the agent can squander significant time in task-irrelevant regions that require careful movements to escape and may overexplore, never returning from indefinite exploration. Recent unsupervised model-based RL (MBRL) approaches (Mendonca et al., 2021; Hu et al., 2023) in the episodic setting have shown sophisticated exploration, high data-efficiency and promising results in long-horizon tasks. This prompts the question: *would MBRL agents excel in the reset-free RL setting?*

As an initial attempt, we first evaluate an unsupervised MBRL agent, in a reset-free Ant locomotion task. The ant is reset to the center of a rectangular arena, and is tasked with navigating to the upper right corner. The agent is reset only once at the start of training. The evaluation is episodic - the agent is reset at the start of each evaluation episode.

For the MBRL agent, we use PEG (Hu et al., 2023), which was developed to solve hard exploration tasks in the episodic setting. As seen in Figure 1, PEG, with minor modifications for the reset-free version, outperforms prior state-of-the-art, model-free agent, IBC (Kim et al., 2023), tailored for the reset-free setting.

In Figure 1, we plot state visitation heatmaps of the agents, where lighter colors correspond to more visitations. The oracle agent, with access to resets, explores the the "task-relevant" area between the initial and top right corner, which is ideal for training a policy that succeeds in episodic evaluation. IBC's heatmap (bottom) shows that it fails to explore effectively, never encountering the goal states in the top right region. In contrast, PEG exhaustively explores the entire space, as seen through its uniform heatmap. This results in an overexploration problem - PEG may devote considerable time on finding irrelevant states rather than concentrating on the task-relevant region of the task. This leads us to ask: **how can MBRL agents acquire more task-relevant data in the reset-free setting to improve its performance?**

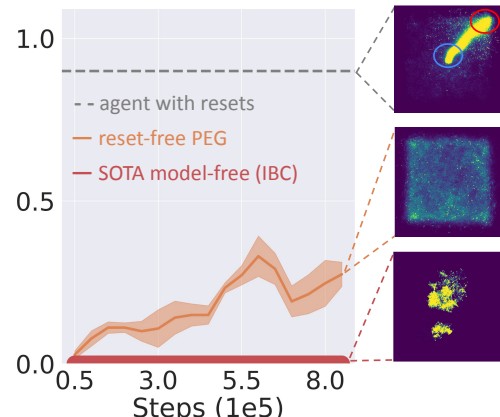

Figure 1: Performance and collected data of different agents on the reset-free Ant locomotion task.

We propose **Mo**del-based, **Re**set-**Free** (MoReFree), which improves two key mechanisms in model-based RL, exploration and policy optimization, to better handle reset-free training. Following the top row of Figure 2: to gather task-relevant data without resets, we define a training curriculum that alternates between temporally extended phases of task solving, resetting, and exploration. Next, as seen in the bottom row of Figure 2, we bias the policy training within the world model towards achieving task-relevant goals such as reaching initial states and evaluation states.

Our key contributions are as follows: **(1)** We demonstrate the viability of using model-based agents with strong exploration abilities for the reset-free setting as well as their inherent limitations. We address such limitations through the MoReFree framework which focuses exploration and policy optimization on task-relevant states. **(2)** We evaluate the adapted reset-free version of MBRL baseline and MoReFree against state-of-the-art reset-free methods in 8 challenging reset-free tasks ranging from manipulation to locomotion. Notably, both model-based approaches outperform prior state-of-the-art baselines in 7/8 tasks in final performance and data efficiency, all the while requiring less supervision (e.g. environmental reward or demonstrations). MoReFree outperforms the model-based baseline in the 3 hardest tasks. **(3)** We perform in-depth analysis of the MoReFree and baselines behaviors, and show that MoReFree explores the state space thoroughly while

retaining high visitation counts in the task-relevant regions. Our ablations show that the performance gains of MoReFree come from the proposed design choices and justify the approach.

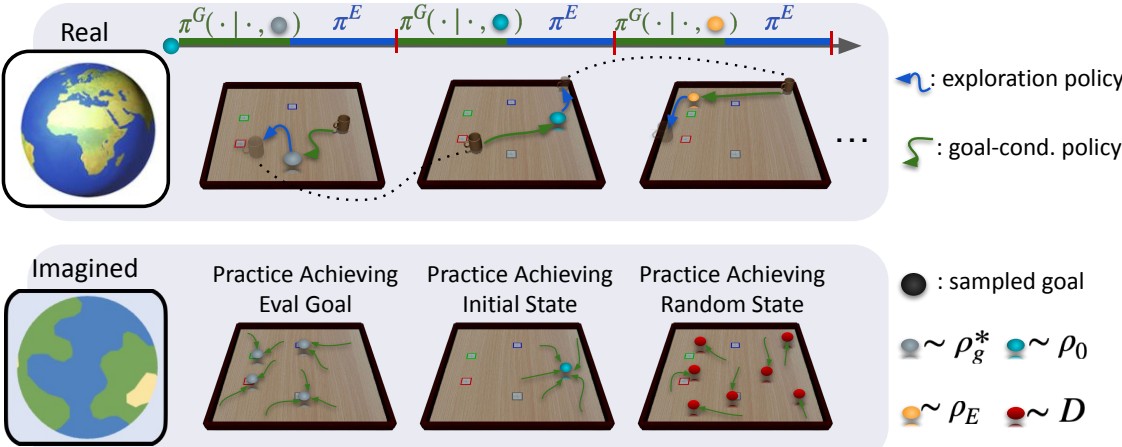

Figure 2: MoReFree is a model-based RL agent for solving reset-free tasks. **Top row:** MoReFree strikes a balance between exploring unseen states and practicing optimal behavior in task-relevant regions by directing the goal-conditioned policy to achieve evaluation states, initial state states (emulating a reset), and exploratory goals. **Bottom row:** MoReFree focuses the goal-conditioned policy training inside the world model on achieving evaluation states, initial states, and random replay buffer states to better prepare the policy for the aforementioned exploration scheme.

## 2  Related Work

**Reset-free RL:** There is a growing interest in researching reinforcement learning methods that can effectively address the complexities of reset-free training. Sharma et al. (2021b) proposes a reset-free RL benchmark (EARL) and finds that standard RL methods like SAC (Haarnoja et al., 2018) fail catastrophically in EARL. Multiple approaches have been proposed to address reset-free training, which we now summarize. One approach is to add an additional reset policy, to bring the agent back to suitable states for learning (Eysenbach et al., 2017; Kim et al., 2022; Sharma et al., 2021a; 2022; Kim et al., 2023). LNT (Eysenbach et al., 2017) and Kim et al. (2022) train a reset policy to bring the agent back to initial state distribution, supervised by dense rewards and demonstrations respectively. MEDAL (Sharma et al., 2022; 2023), train a goal-conditioned reset policy and direct it to reset goal states from demonstrations. IBC (Kim et al., 2023) defines a curriculum for both task and reset policies without requiring demonstrations. VaPRL (Sharma et al., 2021a) trains a single goal-conditioned policy to reach high value states close to the initial states. Instead of guiding the agent back to familiar states, R3L (Zhu et al., 2020) and Xu et al. (2020) learn to reset the policy to diverse initial states, resulting in a policy that is more robust to variations in starting states. However, such methods are limited to tasks where exploration is unchallenging. The vast majority of reset-free approaches are model-free, with a few exceptions (Lu et al., 2020b;a). Other works (Gupta et al., 2021; Smith et al., 2019) model the reset-free RL training process as a multi-task RL problem and require careful definition of the task distribution such that the tasks reset each other.

**Goal-conditioned Exploration:** A common theme running through the aforementioned work is the instantiation of a curriculum, often through commanding goal-conditioned policies, to keep the agent in task-relevant portions of the environment while exploring. Closely related is the subfield of goal-conditioned exploration in RL, where a goal-conditioned agent selects its own goals during training time to generate data. There is a large variety of approaches for goal selection, such as task progress (Baranes & Oudeyer, 2013; Veeriah et al., 2018), intermediate difficulty (Florensa et al., 2018), value disagreement (Zhang et al., 2020), state novelty (Pong et al., 2019; Pitis et al., 2020), world model error (Hu et al., 2023; Sekar et al., 2020), and more. Many goal-conditioned exploration methods use the "Go-Explore" (Ecoffet et al., 2021) strategy, which first selects a goal and runs the goal-conditioned policy ("Go"-phase), and then switches

to an exploration policy for the latter half of the episode ("Explore"-phase). PEG (Hu et al., 2023), which MoReFree uses, extends Go-Explore to the model-based setting, and utilizes the world model to plan states with higher exploration value as goals. However, such methods are not designed for the reset-free RL setting, and may suffer from *over-exploration* of task-irrelevant states.

**Learned Reward Functions:** Instead of requiring the environment to provide a reward function, the agent can learn its own reward function from onboard sensors and data. Given human specified example states, e.g. a goal image, VICE and C-Learning train reward classifiers over examples (Fu et al., 2018; Eysenbach et al., 2021) and agent data. The learned dynamical distance function (Hartikainen et al., 2019) learns to predict the number of actions between pairs of states. The dynamical distance function is used by unsupervised MBRL approaches like LEXA and PEG (Mendonca et al., 2021; Hu et al., 2023) to train the goal-conditioned policy. MoReFree also employs the dynamical distance function as the reward function to eliminate the need of the environmental reward.

Table 1: A conceptual overview of reset-free methods. Existing methods are model-free, and most of them require other forms of supervision (environmental reward or demonstrations or both). In performance, MoReFree improves over reset-free PEG, which significantly outperforms privileged baselines IBC, MEDAL and R3L.

| Approach | MEDAL | IBC | VaPRL | R3L | reset-free PEG | MoReFree |
|---|---|---|---|---|---|---|
| Model-based | ✗ | ✗ | ✗ | ✗ | ✓ | ✓ |
| Demonstrations | ✓ | ✗ | ✓ | ✗ | ✗ | ✗ |
| Environmental reward | ✓ | ✓ | ✓ | ✗ | ✗ | ✗ |

We notice that the majority of all prior work are model-free and may suffer from poor sample efficiency and exploration issues. In contrast, our model-based approaches use world models to efficiently train policies and perform non-trivial goal-conditioned exploration with minimal supervision. See Table 1 for a conceptual comparison between prior work and two model-based methods (MoReFree and reset-free PEG).

## 3 Preliminaries

### 3.1 Reset-free RL

We follow the definition of reset-free RL from EARL (Sharma et al., 2021b), and extend it to the goal-conditioned RL setting. Consider the goal-conditioned Markov decision process (MDP) $\mathcal{M} = (\mathcal{S}, \mathcal{G}, \mathcal{A}, p, r, \rho_0, \rho_{g^*}, \gamma)$. At each time step $t$ in the state $s_t \in \mathcal{S}$, a goal-conditioned policy $\pi(\cdot|s_t, g)$ under the goal command $g \in \mathcal{G}$ selects an action $a_t \in \mathcal{A}$ and transitions to the next state $s_{t+1}$ with the probability $p(s_{t+1}|s_t, a_t)$, and gets a reward $r(s_t, a_t, g)$. $\rho_0$ is the initial state distribution, $\rho_{g^*}$ is the evaluation goal distribution, and $\gamma$ is the discount factor.

The learning algorithm $\mathbb{A}$ is defined: $\{s_i, a_i, s_{i+1}\}_{i=0}^{t-1} \mapsto (a_t, \pi_t)$, which maps the transitions collected until the time step $t$ to the action $a_t$ the agent should take in the non-episodic training and the best guess $\pi_t$ of the optimal policy $\pi^*$ on the evaluation goal distribution ($\rho_{g^*}$). In reset-free training the agent will only be reset to the initial state $s_0 \sim \rho_0$ a few times. The evaluation of agents is still episodic. The agent always starts from $s_0 \sim \rho_0$, and is asked to achieve $g \sim \rho_{g^*}$. The evaluation objective for a policy $\pi$ is:

$$J(\pi) = \mathbb{E}_{s_0 \sim \rho_0, g \sim \rho_{g^*}, a_j \sim \pi(\cdot|s_j, g), s_{j+1} \sim p(\cdot|s_j, a_j)}[\sum_{j=0}^{T} \gamma^j r(s_j, a_j, g)], \tag{1}$$

where $T$ is the total time steps during the evaluation. The goal of algorithm $\mathbb{A}$ during the reset-free training is to minimize the performance difference $\mathbb{D}(\mathbb{A})$ of the current policy $\pi_t$ and the optimal policy $\pi^*$:

$$\mathbb{D}(\mathbb{A}) = \sum_{t=0}^{\infty} (J(\pi^*) - J(\pi_t)). \tag{2}$$

In summary, the algorithm $\mathbb{A}$ should output an action $a_t$ that the agent should take in the non-episodic data collection and a policy $\pi_t$ that can maximize $J(\pi_t)$ at every time step $t$ based on all previously collected data.

### 3.2 Model-based RL setup

Recent goal-conditioned MBRL approaches like LEXA (Mendonca et al., 2021) and PEG (Hu et al., 2023) train goal-conditioned policies purely using synthetic data generated by learned world models. Their robust exploration demonstrates significant success in solving long-horizon goal-conditioned tasks. In the reset-free setting, strong exploration is crucial, as the agent can no longer depend on episodic resets to bring it back to task-relevant areas if it gets stuck. Therefore, we select PEG as the backbone MBRL agent for its strong exploration abilities and sample efficiency.

PEG (Hu et al., 2023) is a model-based Go-Explore framework that extends LEXA (Mendonca et al., 2021), an unsupervised goal-conditioned variant of DreamerV2 (Hafner et al., 2020). The following components are parameterized by $\theta$ and learned:

$$
\begin{aligned}
&\text{world model: } \widehat{\mathcal{T}}_\theta(s_t|s_{t-1}, a_{t-1}) \\
&\text{goal conditioned policy: } \pi_\theta^G(a_t|s_t, g) \qquad \text{goal conditioned value: } V_\theta^G(s_t, g) \\
&\text{exploration policy: } \pi_\theta^E(a_t|s_t) \qquad\qquad\quad \text{exploration value: } V_\theta^E(s_t)
\end{aligned}
\tag{3}
$$

The world model is a recurrent state-space model (RSSM) which is trained to predict future states and is used as a learned simulator to train the policies and value functions. The goal-conditioned policy $\pi_\theta^G$ is trained to reach random states sampled from the replay buffer. The exploration policy $\pi_\theta^E$ is trained on an intrinsic motivation reward that rewards world model error, expressed through the variance of an ensemble (Sekar et al., 2020)(see Appendix C.3 for more details). Both policies are trained on simulated trajectory rollouts in the world model.

▶ **Self-supervised Goal-reaching Reward Function:** Rather than assuming access to the environmental reward, PEG learns its own reward function. PEG uses a dynamical distance function (Hartikainen et al., 2019) as the reward function within world models, which predicts the number of actions between a start and goal state. The distance function is trained on random state pairs from imaginary rollouts of $\pi_\theta^G$. $\pi_\theta^G$ is then trained to minimize the dynamical distance between its states and commanded goal state in imagination. See Appendix C.1 for more details.

▶ **Phased Exploration via Go-Explore:** For data-collection, PEG employs the Go-Explore strategy. In the "Go"-phase, a goal is sampled from some goal distribution $\rho$. The goal-conditioned policy, conditioned on the goal is run for some time horizon $H_G$, resulting in trajectory $\tau_g$.

Then, in the "Explore"-phase, starting from the last state in the "Go"-phase, the exploration policy is run for $H_E$ steps, resulting in $\tau_e$. The interleaving of goal-conditioned behavior with exploratory behavior results in more directed exploration and informative data. This in turn improves accuracy of the world model, and the policies that train inside the world model. See Algorithm 1 and Algorithm 2 for pseudocode. The choice of goal distribution $\rho$ is important for Go-Explore. In easier tasks, the evaluation goal distribution $\rho_{g^*}$ may be sufficient. But in longer-horizon tasks, evaluation goals may be too hard to achieve. Instead, intermediate goals from an exploratory goal distribution $\rho_E$ can help the agent explore. We choose PEG, which generates goals by planning through the world model to maximize exploration value (see Appendix C.2 for details).

---

**Algorithm 1** Go-Explore

1: **Input:** $g, \pi_\theta^G, \pi_\theta^E$
2: $\tau_g \leftarrow \{\}; \tau_e \leftarrow \{\}$
3: **for** $t = 1$ to $H_G$ **do**
4: $\quad a_t \sim \pi_\theta^G(\cdot \,|s_t, g)$
5: $\quad s_{t+1} \sim \mathcal{T}(\cdot \,|s_t, a_t)$
6: $\quad \tau_g \leftarrow \tau_g \cup \{s_t\}$
7: **end for**
8: **for** $t = 1$ to $H_E$ **do**
9: $\quad a_t \sim \pi_\theta^E(\cdot \,|s_t)$
10: $\quad s_{t+1} \sim \mathcal{T}(\cdot \,|s_t, a_t)$
11: $\quad \tau_e \leftarrow \tau_e \cup \{s_t\}$
12: **end for**
13: **return** $\tau_g, \tau_e$

---

## 4 Method

As motivated in Section 1 and Figure 1, the direct application of PEG to the reset-free setting shows promising performance but suffers from over-exploration of task-irrelevant states. To adapt model-based RL to the reset-free setting, we introduce MoReFree, a model-based approach that improves PEG to handle the lack of resets and overcome the over-exploration problem. MoReFree improves two key mechanisms of MBRL for reset-free training: exploration and policy training.

### 4.1 Back-and-Forth Go-Explore

First, we introduce MoReFree's procedure for collecting new datapoints in the real environment. PEG (Hu et al., 2023) already has strong goal-conditioned exploration abilities, but was developed for solving episodic tasks. Without resets, PEG's Go-Explore procedure can undesirably linger in unfamiliar but task-irrelevant portions of the state space. This generates large amounts of uninformative trajectories, which in turn degrades world model learning and policy optimization.

MoReFree overcomes this by periodically directing the agent to return to the states relevant to the task (i.e. initial and evaluation goals). We call this exploration procedure "Back-and-Forth Go-Explore", where we sample pairs of initial and evaluation goals and ask the agent to cycle back and forth between the goal pairs, periodically interspersed with exploration phases (see Figure 2 top row).

Now, we define the "Back-and-Forth Go-Explore" strategy as seen in Algorithm 3. First, we decide whether to perform initial / evaluation state directed exploration. With probability $\alpha$, we sample goals $(g^*, g_0)$ from $\rho_{g^*}, \rho_0$ respectively. Then, we execute the Go-Explore routine for each goal. We name Go-Explore trajectories conditioned on initial state goals

---

**Algorithm 2** MBRL Backbone

1: **Input:** $\pi_\theta^G$, $\pi_\theta^E$, world model $\widehat{\mathcal{T}}_\theta$, goal distribution $\rho$ (including: exploratory goal distribution $\rho_E$, evaluation goal distribution $\rho_{g^*}$, initial state distribution $\rho_0$)
2: $\mathcal{D} \leftarrow \{\}$
3: **while** within the reset-free horizon **do**
4:      # reset-free PEG
5:      sample a goal $g \sim \rho_E$
6:      $\tau_g, \tau_e \leftarrow$ Go-Explore$(g, \pi_\theta^G, \pi_\theta^E)$
7:      # MoReFree
8:      $\tau_g, \tau_e \leftarrow$ Back-and-Forth Go-Explore$(\pi_\theta^G, \pi_\theta^E, \rho)$
9:      $\mathcal{D} \leftarrow \mathcal{D} \cup \tau_g \cup \tau_e$
10:      update $\widehat{\mathcal{T}}_\theta$ with $\mathcal{D}$
11:      update $\pi_\theta^E$ with $\widehat{\mathcal{T}}_\theta$ in imagination
12:      update $\pi_\theta^G$ with $\widehat{\mathcal{T}}_\theta$ in imagination, cond. on goals $g'$:
13:      $g' \sim \mathcal{D}$
14:      $g' \sim \mathcal{D}$ with $\text{Pr} = 1 - \alpha$, $g' \sim \rho_{g^*}, \rho_0$ with $\text{Pr} = \alpha$
15: **end while**

---

as "Back" trajectories, and Go-Explore trajectories conditioned on evaluation goals as "Forward" trajectories. With probability $1 - \alpha$, we execute exploratory Go-Explore behavior by sampling exploratory goals from PEG. The difference between reset-free PEG and MoReFree can be seen in Algorithm 2, unlike PEG, MoReFree employs the "Back-and-Forth Go-Explore".

By following this exploration strategy, the agent modulates between various Go-Explore strategies, alternating between solving the task by pursuing evaluation goals, resetting the task by pursuing initial states, and exploring unfamiliar regions via exploratory goals.

### 4.2 Learning to Achieve Relevant Goals in Imagination

Next, we describe how MoReFree trains the goal-conditioned policy in the world model. To train $\pi_\theta^G$, MoReFree samples various types of goals and executes $\pi_\theta^G(\cdot \mid \cdot, g)$ inside the world model to generate "imaginary" trajectories. The trajectory data is scored using the learned dynamical distance reward mentioned in Section 3.2 , and the policy is updated to maximize the expected return. This procedure is called imagination (Hafner et al., 2019), and allows the policy to be trained on vast amounts of synthetic trajectories to improve sample efficiency.

First, we choose to sample evaluation goals from $\rho_{g^*}$ since the policy will be evaluated on its evaluation goal-reaching performance. Next, recall that Back-and-Forth Go-Explore procedure also samples initial states from $\rho_0$ as goals for the Go-phase to emulate resetting behavior. Since we would like $\pi_\theta^G$ to succeed in such cases so that the task is reset, we will also sample from $\rho_0$. Finally, we sample random states from the

---

**Algorithm 3** Back-and-Forth Go-Explore

1: **Input:** $\pi_\theta^G$, $\pi_\theta^E$, $\rho_{g^*}$, $\rho_0$, $\rho_E$
2: Generate a random number $r$ in $[0, 1]$
3: **if** $r < \alpha$ **then**
4:      $g^*, g_0 \sim \rho_{g^*}, \rho_0$
5:      $\tau_{g^*}, \tau_e^1 \leftarrow$ Go-Explore$(g^*, \pi_\theta^G, \pi_\theta^E)$
6:      # Continue from the terminal state of the previous Go-Explore.
7:      $\tau_{g_0}, \tau_e^2 \leftarrow$ Go-Explore$(g_0, \pi_\theta^G, \pi_\theta^E)$
8:      $\tau_g \leftarrow \tau_{g^*} \cup \tau_{g_0}$; $\tau_e \leftarrow \tau_e^1 \cup \tau_e^2$
9: **else**
10:      $g \sim \rho_E$
11:      $\tau_g, \tau_e \leftarrow$ Go-Explore$(g, \pi_\theta^G, \pi_\theta^E)$
12: **end if**
13: **return** $\tau_g, \tau_e$

---

replay buffer to increase $\pi_\theta^G$'s ability to reach arbitrary states. The sampling probability for each goal type is set to $\alpha/2, \alpha/2, 1 - \alpha$ respectively. In other words, MoReFree biases the goal-conditioned policy optimization procedure to focus on achieving task-relevant goals (i.e. evaluation and initial states), as they are used during evaluation and goal-conditioned exploration to condition the goal-reaching policy (see Figure 2 bottom row). This leads to additional changes of line 13 in Algorithm 2.

### 4.3 Implementation Details

Our work builds on the top of PEG (Hu et al., 2023), and we use its default hyperparameters for world model, policies, value functions and temporal reward function. We set the length of each phase for Go-Explore $(H_G, H_E)$ to half the evaluation episode length for each task. We set the default value of $\alpha = 0.2$ for all tasks (never tuned). See Appendix B.3 for more details and the supplemental for MoReFree code.

## 5 Experiments

We evaluate three MBRL methods (PEG (Hu et al., 2023), the extension reset-free PEG and our proposed method MoReFree) and four competitive reset-free baselines on eight reset-free tasks. We aim to address the following questions: 1) Do MBRL approaches work well in reset-free tasks in terms of sample efficiency and performance? 2) What limitations arise from running MBRL in the reset-free setting, and does our proposed solution MoReFree address them? 3) What sorts of behavior do MoReFree and baselines exhibit in such tasks, and are our design choices for MoReFree justified?

**Baselines:** All baselines except for R3L are implemented using official codebases, see Appendix B.2 for details.

- **PEG** (Hu et al., 2023) is the original episodic PEG in which exploratory goals are only sampled once at the beginning of each episode (in the reset-free setting, the episode is extremely long). The goal-conditioned policy and the exploration policy are then executed for the first half and second half of the episode, respectively.

- **reset-free PEG** is a straightforward extension of PEG to the reset-free setting. Exploratory goals are sampled every $H_G + H_E$ steps. Then, the goal-conditioned policy is executed for $H_G$ steps followed by the exploration policy being executed for $H_E$ steps.

- **DreamerV2** (Hafner et al., 2020) is a commonly used MBRL method. The goal-conditioned policy is executed for the whole reset-free episode.

- **MEDAL** (Sharma et al., 2022) requires demonstrations and trains two policies, one for returning to demonstration states and another that achieves task goals.

- **IBC** (Kim et al., 2023) is a competitive baseline that outperforms prior reset-free work (e.g. MEDAL, VaPRL) by defining a bidirectional curriculum for the goal-conditioned forward and backwards (i.e. reset) policies trained using the environmental reward.

- **R3L** (Zhu et al., 2020) trains two policies, one for achieving task goals and another that perturbs the agent to novel states. Notably, it is the only baseline that operates without any additional assumptions (i.e. environmental rewards, demonstrations, and resets).

- **Oracle** is SAC (Haarnoja et al., 2018) trained under the episodic setting on the environmental reward.

Note that most baselines enjoy some advantage over two MBRL methods: MEDAL, IBC and Oracle use ground truth environmental reward, while MEDAL also uses demonstrations and Oracle uses resets. See Table 1 for a conceptual comparison between MoReFree and prior work.

**Environments:** We evaluate MoReFree and baselines on eight tasks (see Figure 3). We select five tasks from IBC's evaluation suite (Kim et al., 2023) of six tasks; (PointUMaze, Tabletop, Sawyer Door, Fetch

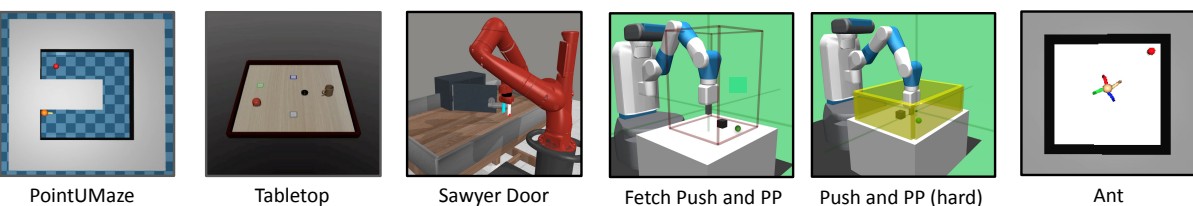

PointUMaze     Tabletop     Sawyer Door     Fetch Push and PP     Push and PP (hard)     Ant

Figure 3: We evaluate MoReFree on eight reset-free tasks ranging from navigation to manipulation. PP is short for Pick&Place.

Push and PP, Fetch Reach is omitted because it is trivially solvable). Next, we increased the complexity of the two hardest tasks from IBC, Fetch Push and Fetch Pick&Place, by extending the size of the workspace, replacing artificial workspace limits (which cause unrealistic jittering behavior near the limits, see the website for videos) with real walls, and evaluating on harder goal states (i.e. Pick&Place goals only in the air rather than including ones on the ground). In addition, we contributed a difficult locomotion task, Ant, which is adapted from the PEG codebase (Hu et al., 2023).

- **PointUMaze**: A point-mass agent navigates a U-shape maze through continuous acceleration commands. During evaluation, the agent starts from the bottom-left corner and is tasked to reach the top-left corner.

- **Tabletop Manipulation**: The agent needs to grab and move the mug to one of the four goal locations. The initial state is always fixed and the goal state is uniformly sampled from four fixed locations.

- **Sawyer Door**: The agent controls a Sawyer robot arm to close the door in an open position. During the reset-free training, it needs to learn to close the door and open the door again to practice. The door is opened to 60 degrees for evaluation.

- **Fetch Push&PP**: The agent commands a Fetch robot arm to push / pick&place the object initialized at the center of the table to goal locations. The environment is taken from IBC's evaluation suite, which modified the original environment from Plappert et al. (2018). To prevent the block from falling off the table, the IBC authors artificially limited the block position with block position constraints.

- **Push&PP(hard)**: Using block position constraints (in Fetch Push&PP) resulted in unrealistic jittering behavior near the limits. To avoid this, we removed the artificial joint constraints and surrounded the table with physical walls. Furthermore, we enable the usage of the grippers (disabled in IBC's version) to permit picking behaviors (i.e. useful for resetting), at the cost of increased action space and exploration difficulty.

- **Ant**: The 4-legged ant agent needs to navigate in a square room to a given goal, which is uniformly located in the top-right corner. The initial state is at the center point with randomness. It is adapted from Hu et al. (2023), with changing the U-shape maze into a square room.

Most methods are run with 5 seeds, and the mean performance and standard error are reported. During the evaluation, the performance on tasks with randomly sampled goals from $\rho_{g^*}$ is measured by averaging over 10 episodes. See Appendix B for more experimental details.

## 5.1 Results

As shown in Fig 4, two reset-free model-based methods (MoReFree and reset-free PEG), without demonstrations or access to environmental reward, outperform other baselines with privileged access to supervision in both final performance and sample efficiency in 7/8 tasks. We observe that the two reset-free MBRL methods learn good behaviors: the pointmass agent hugs the wall of the UMaze to minimize travel time and the Fetch robot deftly pushes and picks up the block into multiple target locations. MoReFree is always competitive

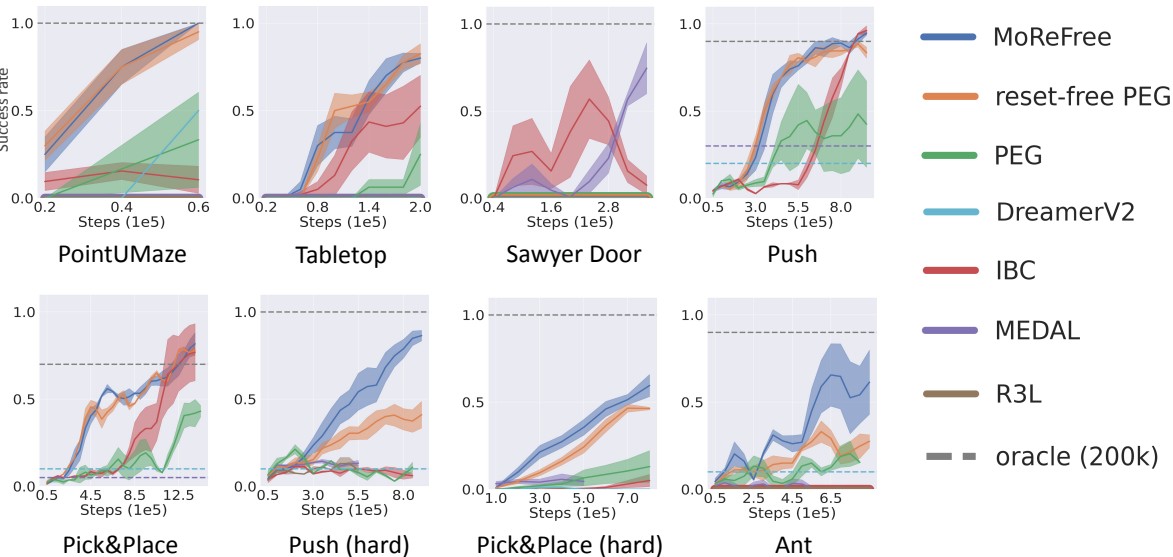

Figure 4: Two reset-free MBRL methods (MoReFree and reset-free PEG) significantly outperform baselines in 7/8 tasks. However, directly applying MBRL methods (PEG and DreamerV2) works poorly. In 4 tasks, only MBRL methods are able to learn meaningful behavior, showcasing MBRL's sample efficiency in the reset-free setting. MoReFree outperforms reset-free PEG in the 3 most difficult tasks.

with or outperforms reset-free PEG, with large gains in the 3 hardest tasks: Push (hard) by 45%, Pick&Place (hard) by 13% and Ant (hard) by 36%. We observe that MoReFree learns non-trivial reset behaviors such as picking and pushing blocks back into the center of the table for the hard variants of the Fetch manipulation tasks. However, the original PEG performs poorly, suggesting that directly applying episodic MBRL methods in a reset-free setting without adaptations yields suboptimal results. See the website for videos of MoReFree and baselines.

In many tasks, the baselines fail to learn at all. We believe this is due the low sample budget, which may be too low for the baselines to fully explore the environment and learn the proper resetting behaviors necessary to train the actual task policy. In Appendix G, we increased the training budget by 3× for the IBC baseline and it still fails, underscoring the difficulty of the tasks and the sample-efficiency gains of MoReFree and MBRL. On the other hand, we noticed that one environment, Sawyer Door, seemed particularly hard for MBRL agents to solve. We hypothesize that the dynamics of the task are hard to model, resulting in performance degradation for model-based approaches (see Appendix F for more analysis).

## 5.2 Analysis

To explain the performance differences between MoReFree and baselines, we closely analyze the exploration behaviors.

**MoReFree focuses on task-relevant states.** In Figure 5 we visualize the state visitation heatmaps of methods in various environments, and also compute the percentage of "task-relevant" states (initial and goal regions, highlighted with white borders). We highlight two trends. First, the heatmaps show that MoReFree and reset-free PEG explore thoroughly while baselines have more myopic exploration patterns, as seen in the Ant heatmaps at the top.

Next, performance differences between reset-free PEG and MoReFree are intuitively explained by the amount of task-relevant data collected by each agent. In easier environments like Push or Pick&Place where both reset-free PEG and MoReFree encounter similar amounts of task-relevant states, the performance is roughly similar between reset-free PEG and MoReFree. But in harder environments (Ant, Push (hard), Pick&Place (hard)) with larger state spaces and more complicated resetting dynamics, MoReFree collects $1.3 - 5\times$ more task-relevant data and has large performance gains over reset-free PEG. By experiencing more task-relevant

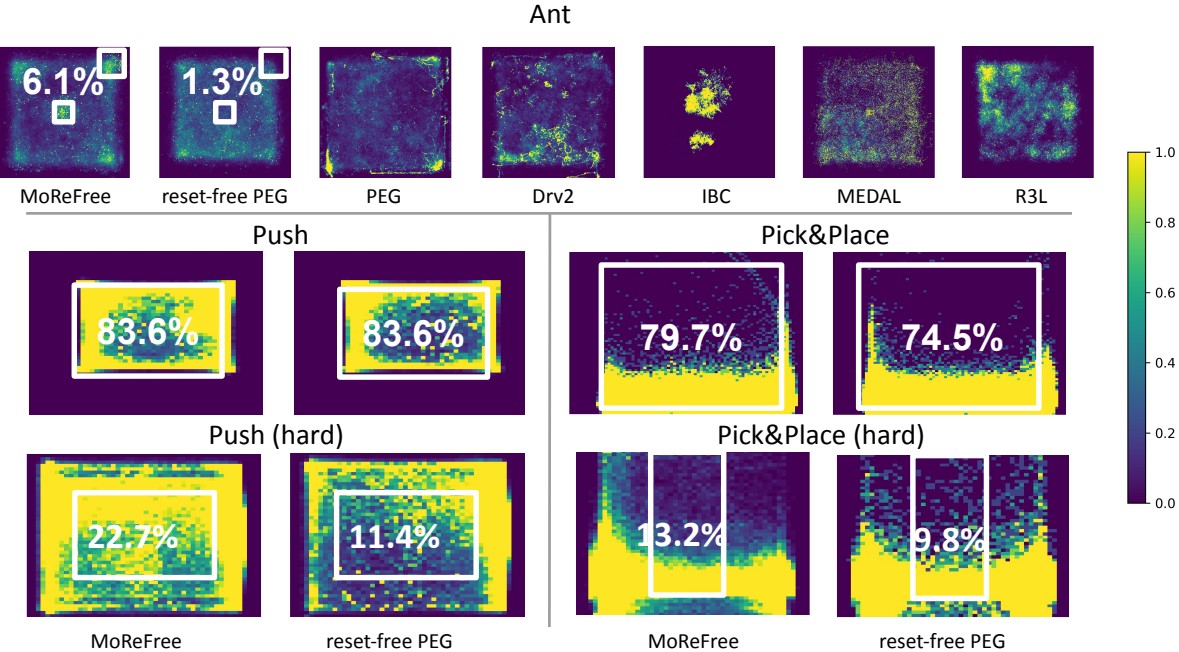

Figure 5: State visitation heatmaps of different agents. White areas are task-relevant states (including initial and goal state distributions) and we overlay the percentages of task-relevant states. reset-free MBRL methods explore more and in harder environments, MoReFree experiences more task-relevant states.

states and training policies on them in imagination, MoReFree policies are more suited towards succeeding at the episodic evaluation criteria. See Appendix D for additional visualizations.

**MoReFree effectively resets.** Next, we investigate the qualitative behavior of MoReFree's Back-and-Forth Go-Explore. To see if "Back" trajectories help free the agent from the sink states, we analyze the replay buffer of MoReFree for the environments, and plot the starting locations of the agent / object up to 100 timesteps before a successful "Back" trajectory is executed in Figure 6. The color intensity of the dots correspond to state density over the last 100 steps (i.e. dark red means the agent / object has rested there for a while). We observe that the starting locations (red dots) of the agent / object are in corners or next to walls in all environments. This suggests that these areas act as sink states, where the agent / object would remain for long and waste time. We observe that MoReFree learns reset behaviors like picking the block out of corners and walls in Fetch Push and Fetch Pick&Place. See detailed videos of the reset behavior on the website.

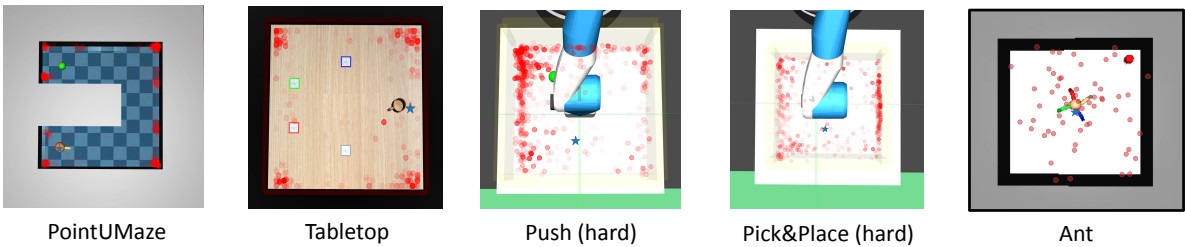

Figure 6: We visualize the start position (red dots) of successful "Back" trajectories of MoReFree's Back-and-Forth Go-Explore, where $\pi_\theta^G$ is directed to reset the environment. The color intensity of the dots correspond to state density over the last 100 steps.

### 5.3 Ablations

To justify our design choices, we ablate the two mechanisms of MoReFree, the back-and-forth exploration and task-relevant goal-conditioned policy training, and plot the results in Figure 7.

First, removing all mechanisms (**MF w/o Explore & Imag.**) reduces to reset-free PEG, and we can see a large gap in performance. Next, **MF with Only Task Goals** sets $\alpha = 1$, which causes an extreme bias towards task-relevant states in the exploration and policy training. This also degrades performance, due to the need for strong exploration in the reset-free setting. Examinations of more values for $\alpha$ can be found in Appendix B.3.

Finally, we isolate individual components of MoRe-Free. First, we disable Back-and-Forth Go-Explore by disallowing the sampling of initial or evaluation goals during Go-Explore. Only exploratory goals are used in Go-Explore for this ablation (named **MF w/o BF-GE**). Next, in **MF w/o Imag.** we turn off the initial / evaluation goal sampling in imagination, so only random replay buffer goals are used to train $\pi_\theta^G$. We see that both variants perform poorly. This is somewhat intuitive, as the two components rely on each other. Having both forms a synergistic cycle where 1) the goal-conditioned policy's optimization is more focused towards reaching initial / goal states, and 2) the exploration is biased towards reaching initial / goal states by using the goal-conditioned policy we just optimized in step 1. If we remove one without the other, then the cycle breaks down. In

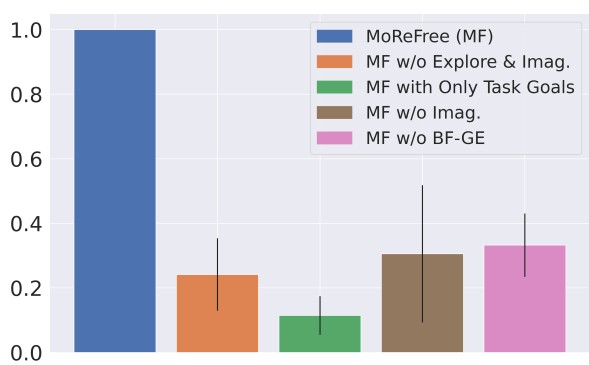

Figure 7: Ablations on 5 variants of MoReFree over 3 hard environments, Push (hard), Pick&Place (hard) and Ant, with normalized final performance.

**MF w/o Imag.**, Back-and-Forth Go-Explore will suffer since $\pi_\theta^G$ trained on random goals cannot reliably reach initial / evaluation goals. In **MF w/o BF-GE**, the exploration strategy will not seek initial / evaluation states, resulting in an inaccurate world model and degraded policy optimization. In summary, the ablations show that MoReFree's design is sound and is the major factor behind its success in the reset-free setting. See Appendix E for details.

## 6 Conclusion and Future Work

As a step towards reset-free training, we adapt model-based methods to the reset-free setting and demonstrate their superior performance. Specifically, we show that with minor modifications, unsupervised MBRL method substantially outperforms the state-of-the-art model-free baselines tailored for the reset-free setting while being more autonomous (requires less supervision like environmental reward or demonstrations). We then identify a limitation of unsupervised MBRL in the reset-free setting (over-exploration on task-irrelevant states), and propose MoReFree to address such limitations by focusing model-based exploration and goal-conditioned policy training on task-relevant states. We conduct a thorough experimental study of MoReFree and baselines over 8 tasks, and show considerable performance gains over the MBRL baseline and prior state-of-the-art reset-free methods.

Despite its overall success, MoReFree is not without limitations. Being a model-based approach, it inherits all associated disadvantages. For example, we believe Sawyer Door is a task where learning the dynamics is harder than learning the policy (see Appendix F), disadvantaging MBRL approaches. Next, MoReFree uses a fixed percentage of task-relevant goals for exploration and imagination, whereas future work could consider an adaptive curriculum. Finally, scaling MoReFree to high-dimensional observations and real-world applications would be natural extensions. We hope MoReFree inspires future efforts in increasing autonomy in RL.

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

## A    Broader Impacts

As we increase the autonomy of RL agents, the possibility of them acting in unexpected ways to maximize reward increases. The unsupervised exploration coupled alongside the learned reward functions further add to the unpredictability; neither mechanisms are very interpretable. As such, we expect research into value alignment, interpretability, and safety to be paramount as autonomy in RL improves.

## B    Experimental Details

### B.1    Environments

**PointUMaze:** The state space is 7D and the action space is 2D. The initial state is $(0,0)$, which located in the bottom-left corner, and noise sampled from $\mathcal{U}(-0.1, 0.1)$ is added when reset. The goal during the evaluation is always located in at the top-left corner of the U-shape maze. The maximum steps during the evaluation is 100. Hard reset will happen after every $2e5$ steps. In the whole training process we performed, it only reset once at the beginning of the training. Taken from the IBC (Kim et al., 2023) paper.

**Tabletop:** The state space is 6D, and the action space is 3D. During the evaluation, four goal locations are sampled in turn, the initial state of the agent is always fixed and located in the center of the table. The maximum steps during the evaluation is 200. Hard reset will happens after every $2e5$ steps. In the whole training process we performed, it only reset once at the beginning of the training. Taken from the EARL (Sharma et al., 2021b) benchmark and also used in the IBC paper.

**Sawyer Door:** The state space is 7D and the action space is 4D. The position of door is initialized to open state (60 degree with noise sampled from $(0, 18)$ degree) and the goal is always to close the door (0 degree). The arm is initialized to a fixed location. Maximum number of steps is 300 for the evaluation. Hard reset will happen after every $2e5$ steps. In the whole training process we performed, it resets twice. Taken from the EARL (Sharma et al., 2021b) benchmark and also used in the IBC paper.

**Fetch Push and Pick&Place:** The state space is 25D and action space is 4D. These are taken from the IBC paper. Authors converted the original Fetch environments to a reversible setting by defining a constraint on the block position. The initial and goal distributions are identical to the original Fetch Push and Pick&Place. More details can be found in the IBC paper.

**Push (hard)**: Different from the original Fetch Push task, in our case walls are added to prevent the block from dropping out of the table. The workspace of the robot arm is also limited. The block is always initialized to a fixed location, and goal distribution during the evaluation is $\mathcal{U}(-0.15, 15)$. Fetch Push used in the IBC paper, the block is limited by joint constraint, which shows unrealistic jittering behaviors near the limits (we observe such phenomenon by running model-based go-explore, the exploration policy prefers to always interact with the block and keep pushing it towards the limit boundary, see videos on our project website [1]). Meanwhile, the gripper is blocked, which makes the task easier. In our case, we release the gripper and it can now open and close again which add two more dimension of the state space. We found it is important to release the gripper in our version of Push task, when the block is in corners, it will need to operate the gripper to drag the block escape from corners. The maximum steps the agent can take in 50 during the evaluation. Hard reset will happen after every $1e5$ steps. In the whole training process we performed, it resets 5 times in total.

**Pick&Place (hard)**: We add walls in the same way as we did for Push (hard). We make it more difficult by only evaluating the agent on goals that are in the air. Then it has to learn to perform picking behavior properly, whereas goals on the ground can just be solved by pushing. The goal will be uniformly sampled from a $5 \times 5 \times 10$ cm cubic area above the table. It has the same observation space, action space, initial state and maximum steps with Fetch Push described above. Hard reset will happens after every $1e5$ steps. In the whole training process we performed, it resets 5 times in total. See the visual difference between our Pick&Place and IBC's in  Figure 3. Since the workspace of the robot is limited within the walls as well in Push (hard) and Pick&Place (hard), when the block gets stuck in corners, the robot needs to precisely move

---

[1]https://yangzhao-666.github.io/morefree

to the corner and bring the block back. In contrast, the robot in IBC's version can move to everywhere, being able to create various circumstance to solve such difficult position.

**Ant:** We adapt the AntMaze task from environments[2] codebase of PEG and change the shape of the maze to square, also change the evaluation goal distribution to be a uniform distribution $\mathcal{U}(2,3)$ for both x and y location, which lies on the top-left corner of the square. The ant is always initialized to the center point $(0, 0)$ of the square to start from, with uniform noise ($\mathcal{U}(-0.1, 0.1)$) added. The state space is 29D and the action space is 8D. The maximum steps for evaluation is 500. Hard reset will happen after every $2e5$ steps. In the whole training process we performed, it reset 4 times in total.

## B.2 Baseline Implementations

**PEG**: We use the official implementation of PEG[3] and only optimize the exploratory goal distribution once at the beginning of each reset-free training episode, i.e. $H_G$ and $H_E$ are set to half of the reset-free episode length.

**reset-free PEG:** We extend the official implementation of PEG[4] to reset-free setting by 1) set $H_G$ and $H_E$ to half of the evaluation episode length; 2) optimizing the goal distribution every $H_G + H_E$ steps; 3) keeping all other hyperparameters the same as MoReFree.

**IBC:** We use the official implementation from authors[5] and keep hyperparameters unchanged.

**DreamerV2**: We use the official implementation of PEG. In order to reduce it to DreamerV2 (Hafner et al., 2020), we remove the exploration policy and only execute goal-conditioned policy for the whole reset-free episode. During imagination training, the goal-conditioned policy is only trained on the evaluation goal distribution.

**MEDAL:** We follow the official implementation of MEDAL[6] and use the deafult setting for experiments. Since MEDAL requires demonstrations, for tasks from EARL benchmark, demonstrations are provided. For other environments, we generate demonstrations by executing the final trained MoReFree to collect data. 30 episodes are generated for each task.

**R3L:** We implement R3L agent by modifying the FBRL agent from MEDAL codebase. The backward policy is replaced by an exploration policy trained using the random network distillation (RND) objective (Burda et al., 2018). The RND implementation we follow is from DI-engine[7].

**Oracle:** This is a episodic SAC agent, we use the implementation from MEDAL codebase and keep all the hyper-parameters unchanged.

**MoReFree:** Our agent is built on the model-based go-explore method PEG (Hu et al., 2023), we extend their codebase by adding back-and-forth goal sampling procedure and training on evaluation initial and goal states in imagination goal-conditioned policy training. See our codebase in the supplemental.

## B.3 Hyperparameters

Train ratio (i.e. Update to Data ratio) is an important hyper-parameter in MBRL. It controls how frequently the agent is trained. Every $n$ steps, a batch of data is sampled from the replay buffer, the world model is trained on the batch, and then policies and value functions are trained in imagination. In all our experiments, we only vary $n$ on different tasks. See the table below for different values on different tasks we used through experiments. MoReFree also introduces a new parameter $\alpha$, which we keep $\alpha = 0.2$ for all tasks and did not tune it at all. All other hyperparameters we keep the same as the original code base.

---

[2]https://github.com/edwhu/mrl
[3]https://github.com/penn-pal-lab/peg
[4]https://github.com/penn-pal-lab/peg
[5]https://github.com/snu-larr/ibc_official
[6]https://github.com/architsharma97/medal
[7]https://opendilab.github.io/DI-engine/12_policies/rnd.html

Table 2: Different train ratio we used for different tasks. We keep all other hyperparameters the same as default ones.

| PointUMaze | 2 | Sawyer Door | 5 | Tabletop | 1 | Fetch Push | 2 |
|---|---|---|---|---|---|---|---|
| Fetch Pick&Place | 2 | Push (hard) | 2 | Pick&Place (hard) | 2 | Ant | 2 |

**Different values for** $\alpha$. We examine different values of $\alpha$ in MoReFree on Fetch Push task, which affects how much MoReFree focuses on task-relevant goals in exploration and imagination. In Figure 8, we see that introducing a moderate amount of task-relevant goals ($\alpha$=0.2, $\alpha$=0.5) results in sensible performance, while too many task-relevant goals ($\alpha$=0.7, $\alpha$=1.0) degrades performance. We use the same value of alpha, 0.2, across all tasks, which showcases MoReFree 's consistency.

### B.4   Results Clarification

In Push and Pick&Place results, we retrieved the final performance of MEDAL directly from the IBC paper (dashed purple lines) and did not have time to run R3L in these two environments. R3L is shown to be a lot worse than MEDAL in the MEDAL paper and performs obviously bad in other tasks shown in Figure 4. In Push (hard) and Pick&Place (hard), we ran R3L and MEDAL with less budget since other methods clearly outperform and their learning curves do not show any evidence for going up.

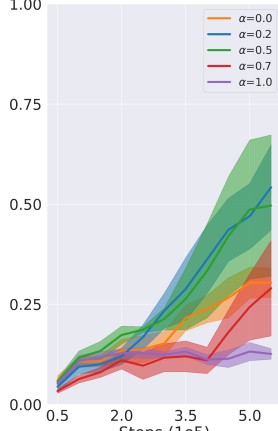

Figure 8: Performance of MoRe-Free with different values of $\alpha$ in Push (hard).

### B.5   Resource Usage

We submit jobs on a cluster with Nvidia 2080, 3090 and A100 GPUs. Our model-based experiments take 1-2 days to finish, and the model-free baselines take half day to one day to run.

## C   Method Details

Here, we provide a more in-depth exposition of self-supervised goal-reaching reward function (Hartikainen et al., 2019) that is used for training the goal-conditioned policy, PEG (Hu et al., 2023) that is used for generating exploratory goal distribution, and P2E (Sekar et al., 2020) that is used for training the exploration policy.

### C.1   Self-supervised Goal-reaching Reward Function

MoReFree does not require environment reward functions, instead it learns a distance function $d_w$ for training the goal-conditioned policy. $d_w$ is trained by sampling pairs of states $s_t, s_{t+k}$ from an imagined rollout of the goal-conditioned policy and predicting the distance $k/H$, where $H$ is the maximum distance equal to the imagination horizon. Then the reward is defined as $r(s_t, g) = -d_w(s_t, g)$.

### C.2   Exploratory Goal Distribution

We use PEG to generate the exploratory goal distribution $\rho_E$ in MoReFree. PEG generates goals that have high exploration potentials. To evaluate a goal $g$, the goal-conditioned policy is rolled out for $K$ trajectories $\tau_k$ within the learned world model. Then the terminal state exploration value for each trajectory with the learned exploration value function $V_\theta^E(s_T^k)$ is estimated, where $s_T^k$ is the last state of the trajectory $\tau_k$. Then the estimates are averaged.

The goal variable $g$ is optimized using model predictive path integral control (MPPI). First, $N$ goal candidates $g$ are sampled from an initial distribution. These candidates are then evaluated as described above. This averaged exploration value acts as the "score" for the goal candidate. Once we have scores for each goal

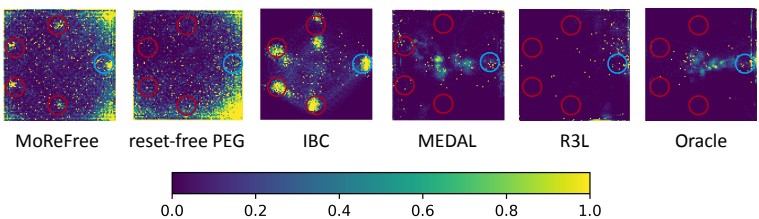

Figure 9: XY state visitation heatmap of the mug in Tabletop of various approaches. MoReFree's heatmap shows high state diversity while retaining high visitation counts near the task-relevant states (red circles are goal states, the blue circle is the initial state). reset-free PEG also shows diverse exploration, but it over-explores the bottom-right corner which is entirely task-irrelevant. IBC's bi-directional curriculum leads the exploration shuttles between the initial state and goal states, but fails to explore well. All other methods fail to explore, visited states mostly cluster in few spots.

candidate, a Gaussian distribution is fit according to the rule:

$$\mu_t = \frac{\sum_{k=0}^{N}(e^{\gamma \cdot V_k})(g_k)}{\sum_{k=0}^{N}(e^{\gamma \cdot V_k})} \tag{4}$$

where $\gamma$ is the reward weight hyperparameter. We then sample candidates from the computed Gaussian, and repeat the process for multiple iterations. After the last iteration, $\rho_E$ is defined as the computed Gaussian.

### C.3 Plan2Explore

The world model $\hat{\mathcal{T}}_\theta$ consists of the following components:

$$
\begin{aligned}
&\text{encoder: } e_t = \text{enc}_\theta(x_t) && \text{posterior: } q_\theta(s_t|s_{t-1}, a_{t-1}, e_t) \\
&\text{dynamics: } p_\theta(s_t|s_{t-1}, a_{t-1}) && \text{decoder: } p_\theta(x_t|s_t)
\end{aligned}
\tag{5}
$$

The model states $s_t$ contain a deterministic component $h_t$ and a stochastic component $z_t$.

P2E is the objective we used to train the exploration policy, and it encourages the agent to visit states that can improve the world model the most. We train an ensemble of 1-step models to predict the next model state from the current model state:

$$\text{Ensemble :} \qquad f(s_t, \theta^k) = \hat{z}_{t+1}^k \qquad \text{for } k = 1...K \tag{6}$$

Then the exploration reward is the variance of the ensemble predictions averaged across dimension of the model state, $r(s_t) = \frac{1}{N}\sum_n \text{Var}_{\{k\}}[f(s_t, \theta_k)]_n$.

## D More Visualizations on Replay Buffer

We visualize the replay buffer of different agents on more tasks. See Figure 9 for XY location of the mug in Tabletop, Figure 11 for XY location data of the agent in PointUMaze, Figure 10 for XZ location of the block in Pick&Place (hard) and Figure 12 for XY location data of the block in Push (hard) and Pick&Place (hard). Overall, we see MoReFree explores the whole state space better. Meanwhile, due to back-and-forth procedure, MoReFree collects more data near initial / goal states, which are important for the evaluation. However, IBC, MEDAL, R3L and Oracle all fail to explore well; their heatmaps are mostly populated with low visitation cells.

## E Detailed Ablations

We report learning curves for each variant agent we ablate in Section 5.3 on every task in Figure 13. Since MoReFree does not learn at all in Saywer Door task, we exclude the ablation for it. In each task, MoReFree

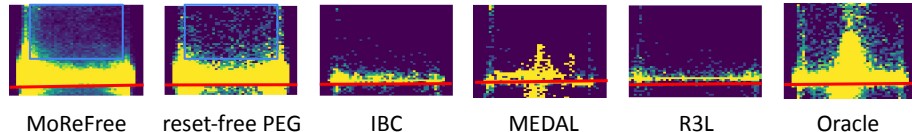

Figure 10: XZ state visitation heatmap of the block in Pick&Place (hard). States above the red line are in the air, which are crucial for solving the picking task. Two MBRL methods collect more data diversely in the air, while other reset-free methods barely pick up the block.

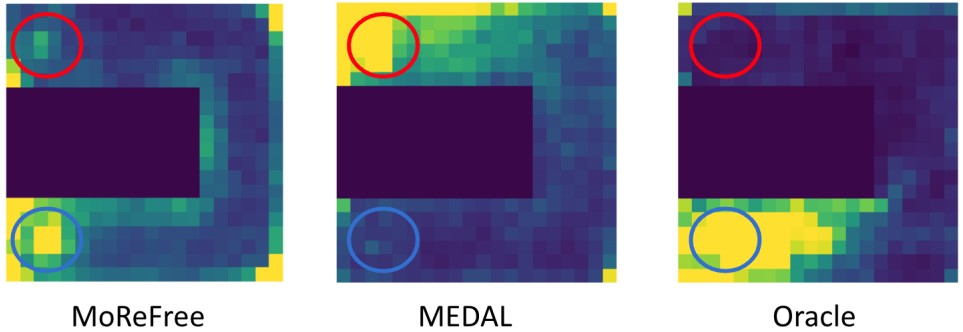

Figure 11: State visitation heatmap on point maze. MoReFree has special focuses on both initial state (blue circles) corner and goal state (red circles), while explore much uniformly. MEDAL collects lots of data near the goal state and little data on the initial state. Both MEDAL and Oracle explore less extensively.

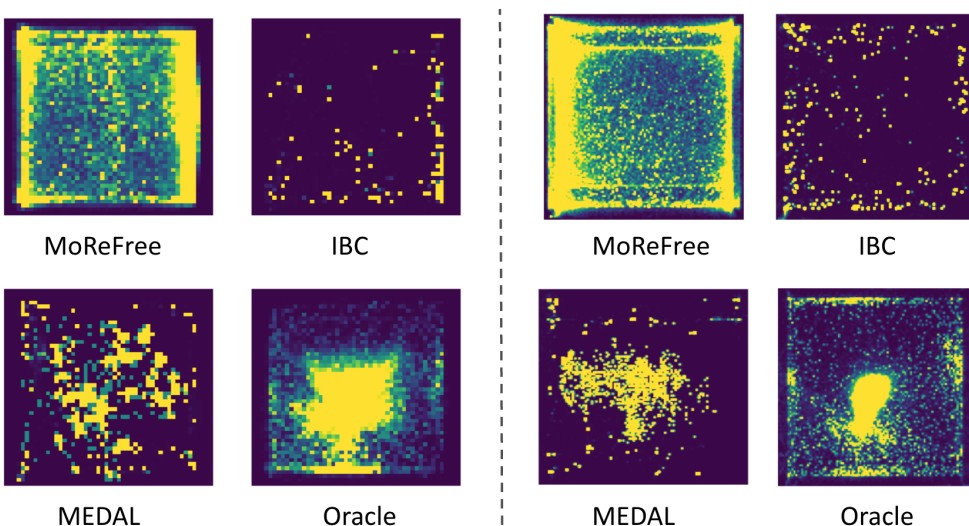

Figure 12: Block state visitation heatmap on Fetch Push (left) and Fetch Pick&Place (right) of different agents. MoReFree better explores the whole state space, while IBC and MEDAL do not have too much interactions with the block, thus lighted areas are scattered everywhere.

is better or on par with all other ablations. Through learning curves, we see different components contribute differently on different tasks.

We further analyze the ablation on PointUMaze as an example by visualizing the replay buffer of different variants, see Figure 14. In the performance on PointUMaze from Figure 13, sampling exploratory goals for data collection is important (MF w/o Explore & Imag. outperforms other ablations). But we see in 14, MF w/o Explore & Imag. does not have focus on the initial / goal state which we care about for the evaluation, which makes it slightly worse than MoReFree. MF with Only Task Goals has a strong preference

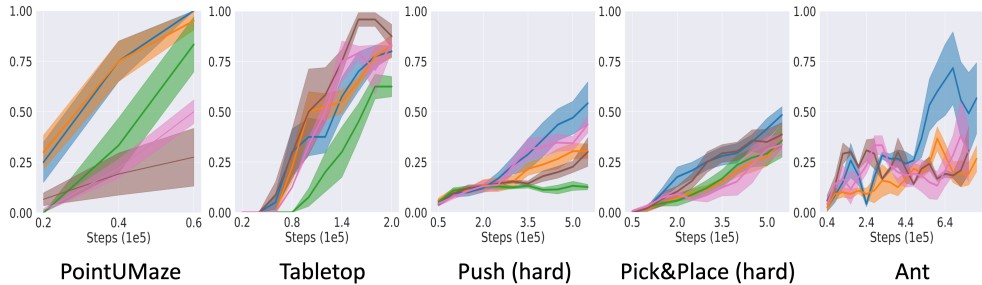

Figure 13: Learning curves of ablation study on 5 tasks. We see different components contribute differently in different tasks. For instance, in Tabletop, **MF w/o Imag.** even performs better than MoReFree, maybe because the whole state space can be explored quickly, then randomly sampling states from the replay buffer as goals for training already has good coverage on evaluation initial / goal states.

on initial / goal state, we think it is because in the later phase of the training when the agent is able to solve the task, it goes back-and-forth consistently to collect data. But in the early phase of the training, it might lack exploration which causes the degraded performance compare with MoReFree. MF w/o Explore and MF w/o Imag. only either go to initial / goal state for data collection and do not practice on it during the imagination training, or practice without really going, which both does not form the positive cycle, and end up with poor performance.

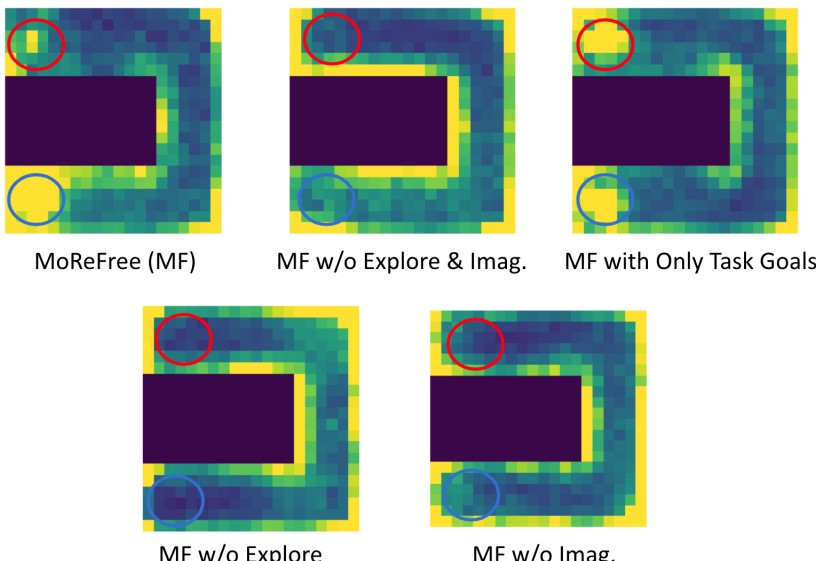

Figure 14: State visitation heatmap on PointUMaze task of all ablations. Red circles are evaluation goal states and blues are initial states. We see MoReFree collect good amount of data near initial / goal states while stronger exploration. MF w/o Explore and MF w/o Imag. could not gather task-relative data, which further causes poor performance.

## F MBRL on Sawyer Door

We investigate why two MBRL methods fail on Sawyer Door tasks. Note that MoReFree is able to solve intermediate goals such as closing the door in some angles, but is unable to solve the original IBC evaluation goal (see website for more videos).

We simplify Sawyer Door task by limiting the movement range of the robot to a box and also having a block holds the door to prevent it from opening it too much, see Figure 15. Although MBRL methods are

trained on the simplified environment, we see learning curves on Sawyer Door are completely flat in Figure 4, compared with other baselines trained on the original task. We wonder why MBRL methods can show the same performance and gain benefits as it does in other environments.

MoReFree and reset-free PEG use DreamerV2 as backbone agents and extend it to reset-free settings. We hypothesize that Dreamer itself, even under the episodic setting with task reward function, would not work well. If that's the case, then MBRL methods in the reset-free setting with self-supervised reward function would almost certainly not work either. For example, if the backbone agent cannot model the dynamics precisely, then policy learning, dynamical distance reward learning, will be degraded.

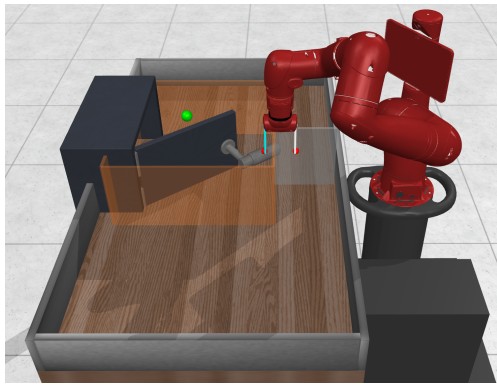

Figure 15: Simplified version of Sawyer Door. Orange walls show the limited workspace for the robot arm, and a grey wall is added to limit the movement of the door. The door can only move to maximum 60 degrees.

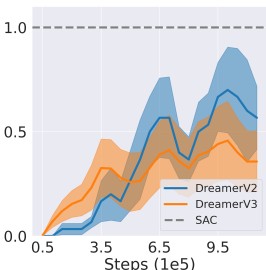

Figure 16: Performance of DreamerV2 and V3 on episodic Sawyer Door task. SAC can solve the task in 200k steps, while after 1 million steps MBRL is still not able to steadily solve the task.

We then run the underlying MBRL backbones under the episodic setting. Figure 16 shows DreamerV2 [8], and Dreamerv3 [9] struggle to solve the task, while model-free method SAC can steadily solve the task after 200k steps. This might be a potential reason that MBRL methods do not work on the more difficult reset-free setting. We hypothesize that the combination of the sparse environmental reward and dynamics of the door result in a hard prediction problem for world modelling approaches. We leave further investigation for the future work.

## G   More Analysis on Fetch Environments

Although IBC gains good final performance in Push and Pick&Place, it starts learning late compared with MBRL methods and fails entirely in our harder versions. We suspect IBC might need more computational budget to start learning in harder tasks. Thus we train IBC with two millions environment steps and results in Figure 17 show that it still fails to solve the harder version of Push.

---

[8] https://github.com/danijar/dreamerv2
[9] https://github.com/danijar/dreamerv3

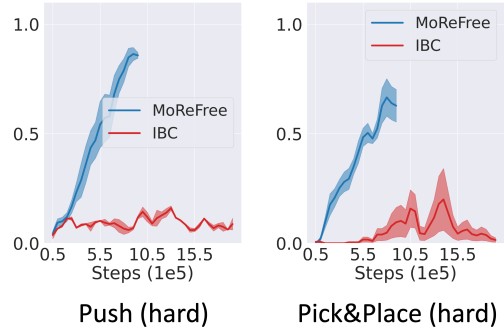

Figure 17: Longer training of IBC in our Fetch tasks, where the state space is larger and artificial constraints are replaced with surrounded walls. IBC still can not learn meaningful behaviors.

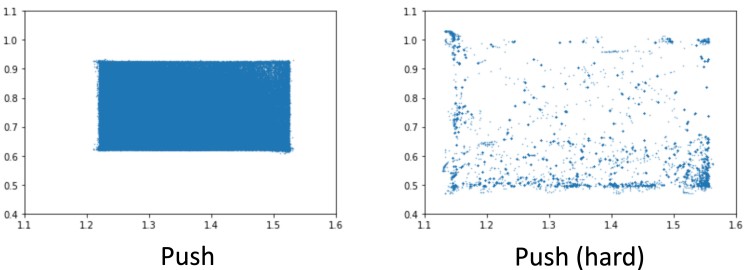

Figure 18: XY location of the block collected by IBC on Push (hard) and its original version (Push). IBC covers the whole state space very well in Push while fails in Push (hard), where the block stays for long time in corners or areas next to walls.

Figure 18 shows 600k data of the obejct (XY view) collected by IBC on our Push (hard) and IBC's Push. We see the block stays in corners or next to walls a lot in Push (hard), while goes everywhere and covers the whole space in IBC's Push, indicating object interaction is more difficult in Push (hard) due to the larger state space, surrounded walls and limited work space. In IBC's Push, the block can bounce back when it hits the limit of joint constraints. However, in Push (hard), the block needs to be explicitly brought back from the corner or walls, requiring more sophisticated behaviors. Meanwhile, larger size of the limited area (our version is $3\times$ larger than IBC's.) also increases the difficulty of the task.

## H   Analysis on R3L

R3L trains two policies, one for reaching the goal and another that brings the agent to novel states. The goal-reaching policy is trained using a learned classifier to classify the goal state and other states. Original R3L takes images as inputs, thus the trained classifier can successfully classify goal images from random state images. In our work, we use low-dimensional state input. Outputs of the trained classifier on the whole state space of PointUMaze is shown in Figure 19. We see that the classifier learns to output higher values for states close to the goal state (red dot) and lower values for states further away. Nonetheless, due to the smoothness of the output scope, states near the initial state (blue circle) that are numerically closer but spatially further to the goal state also have higher values. R3L agent trained using such reward function will always tend to follow states with higher values to the corner instead of going forward. See the website for more videos. These trained reward functions are misleading for learning reasonable policies which result in poor performance we see in Figure 4.

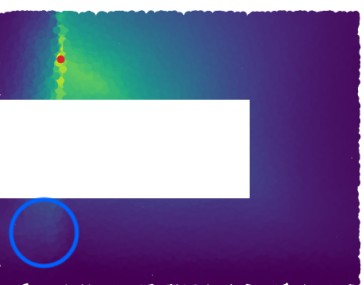

Figure 19: Outputs of the learned classifier on the whole state space. Due to the smoothness of the output scope, states near the initial state (blue circle) also have higher values.

