# OpenReview forum: "Reset-free Reinforcement Learning with World Models"
_TMLR — Accepted by TMLR_

### Review · Reviewer_ybee · 2024-11-12

**Summary Of Contributions:**

This paper introduces MoReFree (Model-based Reset-Free), an RL agent designed for reset-free tasks. This paper first demonstrates the potential of model-based RL in reset-free settings, by adapting an existing MBRL method (PEG) and showing it outperforms prior state-of-the-art reset-free methods on a locomotion task.  Then, it propose MoReFree to address the over-exploration issue in PEG by proposing two key mechanisms: 1) Back-and-Forth Go-Explore; 2) Task-Relevant Goal-Conditioned Policy Training.

The authors evaluate MoReFree and several baselines (including the adapted PEG, IBC, MEDAL, and R3L) on eight reset-free tasks, ranging from manipulation to locomotion.  MoReFree demonstrates good performance and sample efficiency compared to the baselines in many tasks.

Analysis of the agent's behavior reveals that MoReFree explores the state space effectively while maintaining a higher concentration of experience in task-relevant regions.  Ablation studies confirm the importance of both the Back-and-Forth Go-Explore and the task-focused policy training.

The key contributions of this paper are:
-  Demonstrate the viability of using model-based agents with strong exploration abilities for the reset-free setting as well as their inherent limitations.
-  Address such limitations through the MoReFree framework which focuses exploration and policy optimization on task-relevant states.

**Audience:**

Yes

**Claims And Evidence:**

Yes

**Requested Changes:**

See weaknesses.

**Strengths And Weaknesses:**

## Strengths

- This paper points out an important problem of reducing human effort in RL training, which is a significant bottleneck for real-world applications.  The focus on reset-free RL is very relevant to this challenge.

- Combining a tailored exploration strategy (Back-and-Forth Go-Explore) with targeted policy training within a learned world model seems to be a good idea to address the over-exploration problem observed in prior MBRL adaptations for reset-free tasks.

- MoReFree achieves good performance without relying on environment rewards or expert demonstrations, unlike many existing reset-free methods.  This increased autonomy is a significant advantage for practical applications.

- This paper provides a clear analysis of MoReFree's behavior, including visualizations of state visitations and targeted investigations of the "Back" trajectories.  In addition, the ablation studies evaluate the contributions of MoReFree's individual components, providing clear evidence for the effectiveness of the proposed two strategies.


## Weaknesses


- It seems the proposed method exhibit limitations when handling tasks with *irreversible consequences*. While in the Ant task, the agent can freely relocate to achieve different goals, many real-world scenarios do not offer such flexibility. For instance, in table-top manipulation tasks, if a robot kick an object out of the the workspace, learning becomes impossible until the environment is reset by someone else. The core challenge lies in moving between arbitrary current states and goals, which may be infeasible in numerous scenarios. The paper's reported failures in the Sawyer Door task likely stem from similar difficulties. While the proposed method works well in many tasks, I feel its applicability is constrained in environments where actions can lead to irreversible consequences.


- The proposed method requires access to a goal distribution, which limits its applicability to tasks with a single fixed goal. For instance, in tasks like plugging a charger into a socket, where only one specific goal state exists, it remains unclear how the method could be effectively implemented.

- Based on Figure 4, the performance difference between MoReFree and the reset-free PEG baseline appears marginal.

- The current implementation uses a fixed parameter $\alpha$ to control the balance between task-relevant and exploratory goals.  This fixed curriculum might not be optimal for all environments or stages of learning. An adaptive curriculum that adjusts $\alpha$ based on the agent's progress could potentially improve performance.

- While the motivation is rooted in reducing human effort for real-world applications, the experiments are still conducted in simulated environments. Demonstrating the effectiveness of MoReFree on real-world robotic tasks would significantly improve the work.

---

> ### Author Response · Authors · 2024-12-01
> **Author Response (01/12/24)**
>
> Thank you for your review!
>
> > It seems the proposed method exhibit limitations when handling tasks with irreversible consequences. While in the Ant task, the agent can freely relocate to achieve different goals, many real-world scenarios do not offer such flexibility. For instance, in table-top manipulation tasks, if a robot kick an object out of the the workspace, learning becomes impossible until the environment is reset by someone else. The core challenge lies in moving between arbitrary current states and goals, which may be infeasible in numerous scenarios. The paper's reported failures in the Sawyer Door task likely stem from similar difficulties. While the proposed method works well in many tasks, I feel its applicability is constrained in environments where actions can lead to irreversible consequences.
>
> This is indeed a concern, but it is assumed by reset-free RL methods in general that try to learn reset behaviors.  If transitions are irreversible, then some resets will be physically impossible, and any reset-free RL algorithm would not succeed.
>
> We ran DreamerV2 and V3 on the **episodic** Sawyer Door task (see Appendix E and Fig.16), and it turns out they do not work very well, compared to model-free baselines. So the failure in the Sawyer Door is not from the irreversibility property, instead, our intuition is that learning dynamics in such a task is difficult.
>
> > The proposed method requires access to a goal distribution, which limits its applicability to tasks with a single fixed goal. For instance, in tasks like plugging a charger into a socket, where only one specific goal state exists, it remains unclear how the method could be effectively implemented.
>
> For a single goal $g$, that can be viewed as a special case of the goal distribution where $p(g)$ is 1 and 0 for all other goals. It should work in our framework, and in the Tabletop task that we used in the paper, there are four fixed goals, which is similar to the situation you mentioned.
>
> > Based on Figure 4, the performance difference between MoReFree and the reset-free PEG baseline appears marginal.
>
> This is primarily due to the varying difficulty of the tasks. Many of the easier tasks are quickly solved by reset-free PEG or MoReFree, but as the tasks become harder to explore, MoReFree begins to demonstrate superior performance. Meanwhile, MoReFree never performs worse than reset-free PEG.
>
> > The current implementation uses a fixed parameter $\alpha$ to control the balance between task-relevant and exploratory goals. This fixed curriculum might not be optimal for all environments or stages of learning. An adaptive curriculum that adjusts $\alpha$ based on the agent's progress could potentially improve performance.
>
> Thanks for pointing this out. We indeed think that MoReFree would benefit from a dynamically adjusted $\alpha$, and mentioned / left it in the future work.
>
> > While the motivation is rooted in reducing human effort for real-world applications, the experiments are still conducted in simulated environments. Demonstrating the effectiveness of MoReFree on real-world robotic tasks would significantly improve the work.
>
> We agree that examining MoReFree on real-world robotics would be very interesting. But it is out of the scope of this work, so we mentioned it in the “Future Work” and left it for the future work. Please see the updated manuscript.

---

> > ### Comment · Reviewer_ybee · 2024-12-05
> >
> > Thank the authors for the response. As the authors have argued that many of my concerns fall outside the scope of this paper, I have no further comments.

---

### Review · Reviewer_crQe · 2024-11-17

**Summary Of Contributions:**

In this paper, the authors study the setting of reset-free reinforcement learning. They build upon PEG, a Go-Explore style exploration strategy used alongside a goal-conditioned model-based reinforcement learning framework.
- The authors empirically demonstrate that directly applying PEG in a reset-free setting leads to over-exploration.
- They adapt the Go-Explore strategy used in PEG with their Back-and-Forth Go-Explore strategy, which additionally incorporates trajectories with goals sampled from the initial distribution of the environment.
- When updating the goal-conditioned policy by using the learned world model, the authors use a sampling strategy to alternate between sampling goals from the goal, initial, and exploration distribution. By selecting goals from these distributions, the claim is that the goal-conditioned policy learns to reach the goals but also learns to reset the agent to the initial states during the reset-free training procedure.
- In their experiments, the authors demonstrate that the two model-based methods, i.e., reset-free PEG and their method MoReFree, outperform in terms of efficiency model-free baselines in 7 out of 8 tasks.

**Audience:**

Yes

**Broader Impact Concerns:**

No concerns here. The authors include a broader impact paragraph in the appendix.

**Claims And Evidence:**

Yes

**Requested Changes:**

First, I would like to ask the authors to address and clarify the aforementioned weaknesses and open questions/points.

Additionally, to strengthen the paper and make it easier to follow, I recommend the authors to consider the following changes in the submission:

- I recommend that the authors not refer the reader to the two other works (Mendonca et al. (2021) and Hu et al. (2023)) for more details, as done in 3.2. Instead, it is better to include in the paper all the relevant details that we need to know from these two works. Since these details are important to follow and to understand the method, adding these details would make the paper self-contained.

- In Algorithm 2, lines 11 and 12, the policies are updated to maximize some rewards. It would be good to clarify and explain in the paper what are the rewards that the policies try to maximize.

- Clarify further how distribution $\rho_E$ is defined and include it in the paper.

- Include a short description of each environment along with information about each environment's initial and goal location in the main paper.

**Strengths And Weaknesses:**

### Strengths:

- The introduction is well-written, and the motivation for reset-free reinforcement learning is clearly presented.
- The empirical results are comprehensive. The authors compare their method with five different baselines (one model-based and four model-free) in 8 different tasks.
- The authors include an ablation study, where they discuss and justify the use of the different components of their method.


### Weaknesses:

- There is no comparison with another model-based RL method except from the adapted reset-free PEG.
- Although it seems like reset-free PEG leads to over-exploration based on Figure 1, the results in Figure 4 show that reset-free PEG is on par with MoReFree for most tasks – the gap looks convincing mainly for Push (hard) and Ant environment.
- Missing descriptions and information about the environments in the main paper. This information is in the appendix, but a basic description and some information about each environment's initial and goal location would be helpful to be in the main paper.

Regarding weaknesses, I have these additional points for discussion and clarification:

1. In Algorithm 3, line 6, for the Go-Explore process where the goal is sampled from the initial state distribution, it needs to be clarified what is the starting state of that trajectory. Is it the previous goal state or the terminal state of the previous exploration? Could you specify that in the algorithm to make it clear?

2. What happens when $\rho_g*$ and $\rho_0$ are identical to the state space $S$, i.e., every state can be a plausible goal location or initial state. Do the Back-and-Forth Go-Explore and the sampling strategy become trivial in that case? Does that mean that this method makes sense in cases where $\rho_g*$ and $\rho_0$ are distinct subsets of state $S$?

3. It is unclear why, in Algorithm 2, line 5, the goal for reset-free PEG is sampled from $\rho_E$. To me, the equivalent for the reset-free PEG is to be sampled from $\rho_g*$. If the exploration goal distribution $\rho_E$ is different from the evaluation goal distribution $\rho_g*$, reset-free PEG never observes the evaluation goal distribution as MoReFree does. It would be nice if the authors could clarify this point.

4. Could you explain how the choice of the task relevant states is being made in order to get the percentages in Figure 5 (section 5.2 analysis)?

---

> ### Author Response · Authors · 2024-12-01
> **Author Response Part 1 of 2 (01/12/24)**
>
> Thank you for your review!
>
> > There is no comparison with another model-based RL method except from the adapted reset-free PEG.
>
> We now added another model-based baseline, DreamerV2, which performs poorly in reset-free tasks. Please see the updated manuscript.
>
> > Although it seems like reset-free PEG leads to over-exploration based on Figure 1, the results in Figure 4 show that reset-free PEG is on par with MoReFree for most tasks – the gap looks convincing mainly for Push (hard) and Ant environment.
>
> This is primarily due to the varying difficulty of the tasks. Many of the easier tasks are quickly solved by reset-free PEG or MoReFree, but as the tasks become harder to explore, MoReFree begins to demonstrate superior performance. Meanwhile, MoReFree never performs worse than reset-free PEG.
>
> We aim to highlight two key points:
>
> 1. MBRL excels in reset-free settings: Both reset-free PEG and MoReFree outperform all model-free baselines in these scenarios.
> 2. MoReFree surpasses reset-free PEG in more challenging tasks: This is particularly evident in the three harder tasks we evaluated.
>
> > Missing descriptions and information about the environments in the main paper. This information is in the appendix, but a basic description and some information about each environment's initial and goal location would be helpful to be in the main paper.
>
> We now added information about environments in the main paper. Please see the updated manuscript.
>
> > In Algorithm 3, line 6, for the Go-Explore process where the goal is sampled from the initial state distribution, it needs to be clarified what is the starting state of that trajectory. Is it the previous goal state or the terminal state of the previous exploration? Could you specify that in the algorithm to make it clear?
>
> It is the terminal state of the previous exploration process. We updated the Algorithm 3 and please see the updated manuscript.
>
> > What happens when $\rho_{g*}$ and $\rho_0$ are identical to the state space $S$, i.e., every state can be a plausible goal location or initial state. Do the Back-and-Forth Go-Explore and the sampling strategy become trivial in that case? Does that mean that this method makes sense in cases where $\rho_{g*}$ and $\rho_0$  are distinct subsets of state $S$?
>
> If $\rho_{g*}$ and $\rho_0$ become identical to the whole state space, then indeed, we would expect MoReFree to have similar results with reset-free PEG. The evidence can be found in Fig.5, in Push and Pick&Place where the task-relevant states largely overlaps with the state space $S$, reset-free PEG and MoReFree have similar coverage on task-relevant states, resulting in similar performance in Fig.4 as well.
>
> However, MoReFree proves more effective in scenarios where $\rho_{g*}$ and $\rho_0$ are distinct subsets of the state space $S$. For example, in tasks such as Push (hard), Pick & Place (hard), and Ant, where $\rho_{g*}$ and $\rho_0$​ are clearly distinct, MoReFree significantly outperforms reset-free PEG.
>
> > It is unclear why, in Algorithm 2, line 5, the goal for reset-free PEG is sampled from $\rho_E$. To me, the equivalent for the reset-free PEG is to be sampled from $\rho_{g*}$. If the exploration goal distribution $\rho_E$ is different from the evaluation goal distribution  $\rho_{g*}$., reset-free PEG never observes the evaluation goal distribution as MoReFree does. It would be nice if the authors could clarify this point.
>
> The core idea of PEG is to only sample goals from an “exploratory” goal distribution, leading to better exploration performance. We strictly follow the original idea of the baseline, thus only sampling goals from the “exploratory” goal distribution.
>
> > Could you explain how the choice of the task relevant states is being made in order to get the percentages in Figure 5 (section 5.2 analysis)?
>
> Task-relevant states include both initial state distribution and the evaluation goal state distribution.

---

> > ### Author Response · Authors · 2024-12-01
> > **Author Response Part 2 of 2 (01/12/24)**
> >
> > > I recommend that the authors not refer the reader to the two other works (Mendonca et al. (2021) and Hu et al. (2023)) for more details, as done in 3.2. Instead, it is better to include in the paper all the relevant details that we need to know from these two works. Since these details are important to follow and to understand the method, adding these details would make the paper self-contained.
> >
> > We now added the details of these two works in the Appendix C. Please see the updated manuscript.
> >
> > > In Algorithm 2, lines 11 and 12, the policies are updated to maximize some rewards. It would be good to clarify and explain in the paper what are the rewards that the policies try to maximize.
> >
> > We added details of two reward functions in Appendix C.1 and C.3. Please see the updated manuscript.
> >
> > > Clarify further how distribution $\rho_E$ is defined and include it in the paper.
> >
> > We now added details of distribution $\rho_E$ in Appendix C.2. Please see the updated manuscript.
> >
> > > Include a short description of each environment along with information about each environment's initial and goal location in the main paper.
> >
> > We now added information about environments in the main paper. Please see the updated manuscript.

---

### Review · Reviewer_76Wb · 2024-11-22

**Summary Of Contributions:**

The authors examine the role of world models in reset-free reinforcement learning scenarios, identify their limitations, propose a new algorithm to mitigate those limitations, evaluate model based as well as model free baselines on a number of challenging benchmarks, and analyse the algorithms' performances by inspecting the resulting state space visitations.

**Audience:**

Yes

**Broader Impact Concerns:**

-

**Claims And Evidence:**

Yes

**Requested Changes:**

See comments above, please fix fig.2 / add an explanation.

**Strengths And Weaknesses:**

I thank the authors for their great work. The paper is well written and much effort is put into explaining things clearly. The work examines the reset-free RL scenario, which is important for many real-world applications, so I expect it to be impactful. After analysing prior methods either specifically designed for the scenario or adapted for it, the authors propose a novel algorithm which changes both exploration with the new Back-and-Forth Go-Explore methodology, as well as the policy training, by changing the goals that are sampled to train the goal conditioned policy to be biased towards task relevant states. The resulting empirical evaluation demonstrates the improvements over relevant baselines and the analysis & ablations show that it is indeed the changes proposed by the authors that make the difference.

I do not see any major weaknesses of the paper - one could of course include more baselines or environments to make the empirical evaluation even more compelling, however the amount of evidence presented here is enough to make a point.

I have a couple of minor comments / questions:
- for reviewing, it would be great to have line numbers
- What confused me a little was the following sentence: "In reset-free training the agent will only be reset to the initial state s0 ∼ ρ0 a few times". The name "reset-free" kind of implies that this resetting happens 0 times, could elaborate this further? How many times do you reset during training & how often do "normal" algorithms reset?
- In fig. 2 I am missing an explanation for the purple dot / state (bottom row in the middle) - since it says initial state, I would imagine the cooler should be turquoise?
- Typo: "Their robust exploration demonstrate significant ..." (demonstrates)
- Typo: "conduct a through" (thorough)
- One thing I was wondering while reading: You assume to have access to the goal & start state distributions p_g & p_0 so that you can sample them during policy training. I'm wondering whether that is a common & justified assumption in the reset-free setting? In the normal case one could at least get the starting distribution by sampling from the reset function, but we cannot in this case. Also the goal distribution is not directly visible maybe in some scenarios. Could you elaborate a little? Why do you e.g. not use environment reward but do use those two pieces from the original env?

---

> ### Author Response · Authors · 2024-12-01
> **Author Response (01/12/24)**
>
> Thank you for your review!
>
> > What confused me a little was the following sentence: "In reset-free training the agent will only be reset to the initial state s0 ∼ ρ0 a few times". The name "reset-free" kind of implies that this resetting happens 0 times, could elaborate this further? How many times do you reset during training & how often do "normal" algorithms reset?
>
> The agent/environment is reset between 1 to 5 times during the entire reset-free training process, depending on the task. In contrast, a "normal" algorithm typically resets once every 100–500 steps, depending on the task. More detailed information about the frequency of resets can be found in Appendix B.1. Our approach and all previous reset-free works follow the settings of the EARL benchmark, where the agent/environment undergoes limited resets during reset-free training.
>
> > In fig. 2 I am missing an explanation for the purple dot / state (bottom row in the middle) - since it says initial state, I would imagine the cooler should be turquoise?
>
> Thanks for pointing it out. We made a mistake there, and it indeed should be turquoise. We corrected it and please see the updated manuscript.
>
> > Typo: "Their robust exploration demonstrate significant ..." (demonstrates). Typo: "conduct a through" (thorough)
>
> Thanks for pointing these out. We have fixed these typos and please see the updated manuscript.
>
> > One thing I was wondering while reading: You assume to have access to the goal & start state distributions p_g & p_0 so that you can sample them during policy training. I'm wondering whether that is a common & justified assumption in the reset-free setting? In the normal case one could at least get the starting distribution by sampling from the reset function, but we cannot in this case. Also the goal distribution is not directly visible maybe in some scenarios. Could you elaborate a little? Why do you e.g. not use environment reward but do use those two pieces from the original env?
>
> Yes, this is a common assumption in previous reset-free research. It assumes that while the agent/environment cannot be directly reset to the initial state distribution, knowledge of this distribution is available. For instance, during the training of a robotic arm, although the arm cannot be physically reset to a specific initial state, it is beneficial to know the target state and train the arm to return to it.
>
> The advantage of not relying on an environment reward function comes from the underlying model-based approach. By learning a world model, a reward function can be derived using data generated by this model. We leveraged this advantage in our methodology.

---

### Decision · Action_Editor_Ufr3 · 2025-02-19

**Recommendation:** Accept as is

**Comment:**

The paper investigates the role that world models play in the context of reset-free reinforcement learning (RL). The paper identifies the limitations of existing Go-Explore-like exploration (namely PEG) and propose changes to both the exploration strategy (namely, Back-and-Forth Go-Explore that utilizes trajectories with goals sampled from the environment) as well as to policy training (by using a sampling different goal types) in an effort to address some of these limitations.

The reviewers are in agreement regarding the paper's key strengths as well as some of the questions/concerns with the paper as initially submitted. Among the strengths, all three reviewers agree that the paper addresses an important problem (namely, reset-free RL) that is of interest to many in the community. Additionally, the reviewers emphasize the comprehensive nature of the experimental evaluation and the value of the ablation study, which provides evidence of the benefits of the proposed changes to the exploration strategy and policy training. The reviewers similarly agree that the paper is well written and that the presentation is clear and easy to follow. Some reviewers questioned the seemingly small gain in improvement over the PEG baseline, while at least one reviewer suggested the importance of comparing to an additional model-based RL method other than the adapted reset-free PEG baseline. As part of the rebuttal process, the authors included an additional comparison to DreamerV2. Meanwhile, the authors effectively addressed other questions and concerns that the reviewers raised, which were acknowledged in comments to the AE.

**Audience:**

The paper's focus on reset-free reinforcement learning is highly relevant for many real-world applications. Together with the high quality of the writing and the clarity of the presentation, the paper will be of interest to many in the community.

**Claims And Evidence:**

As the reviewers point out, with the addition of the DreamerV2 baseline, the paper provides a comprehensive experimental evaluation of the proposed MoReFree framework. The results demonstrate improvements over suitable baselines, while the ablation study reveals the benefits of the proposed changes. Thus, the paper adequately supports the claims being made.